# LOMIT: Local Mask-based Image-to-Image Translation via Pixel-wise Highway Adaptive Instance Normalization

## Abstract

Recently, image-to-image translation has seen a significant success. Among many approaches, image translation based on an exemplar image, which contains the target style information, has been popular, owing to its capability to handle multimodality as well as its suitability for practical use. However, most of the existing methods extract the style information from an entire exemplar and apply it to the entire input image, which introduces excessive image translation in irrelevant image regions. In response, this paper proposes a novel approach that jointly extracts out the local masks of the input image and the exemplar as targeted regions to be involved for image translation. In particular, the main novelty of our model lies in (1) co-segmentation networks for local mask generation and (2) the local mask-based highway adaptive instance normalization technique. We demonstrate the quantitative and the qualitative evaluation results to show the advantages of our proposed approach. Finally, the code is available at `https://github.com/AnonymousIclrAuthor/Highway-Adaptive-Instance-Normalization`.

## 1 Introduction

Unpaired image-to-image translation (or in short, image translation) based on generative adversarial networks (Goodfellow et al., 2014) aims to transform an input image from one domain to another, without using paired data between different domains (Zhu et al., 2017a; Liu et al., 2017; Kim et al., 2017; Choi et al., 2018; Liu et al., 2017). An unpaired setting, however, is inherently multimodal, denoting a single input image can be mapped to multiple different outputs within a target domain. For example, when translating the hair color of a given image into a blonde, the detailed region (e.g., upper vs. lower, and partial vs. entire) and color (e.g., golden, platinum, and silver) may vary.

Previous studies have achieved such multimodal outputs by adding a random noise sampled from a pre-defined prior distribution (Zhu et al., 2017b) or taking a user-selected exemplar image as additional input, which contains the detailed information of an intended target style (Chang et al., 2018). Recent studies (Lin et al., 2018; Ma et al., 2018) including MUNIT (Huang et al., 2018) and DRIT (Lee et al., 2018) combine those two approaches, showing the state-of-the-art performance by separating (i.e., disentangling) content and style information of a given image through two different encoder networks.

However, existing exemplar-based image translation method has several limitations as follows. First, the style information is typically extracted and encoded from the entire region of a given exemplar, thus being potentially noisy due to those regions involved with respect to the target attribute to transfer. Suppose we translate the hair color of an image using an exemplar image. Since the hair color information is available only in the hair region of an image, the style information extracted from the entire region of the exemplar may contain the irrelevant information (e.g., color of the wall and edge pattern of the floor), which should not be reflected in the intended image translation.

On the other hand, the extracted style is then applied to the entire region of the target image, even though particular regions should be kept as it is. Due to this limitation, some of the previous approaches (Huang et al., 2018; Lee et al., 2018) often distort irrelevant regions of an input image such as the background.

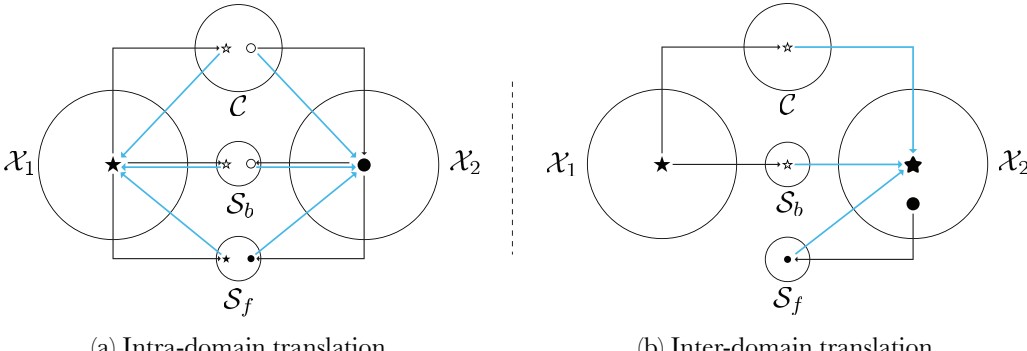

(a) Intra-domain translation            (b) Inter-domain translation

Figure 1: Image translation settings. (a) Each domain $\mathcal{X}_i$ is defined as the subset of data that shares a particular attribute. An image from each domain $\mathcal{X}_i$ is decomposed into a content space $\mathcal{C}$, a foreground style space $\mathcal{S}^f$, and a background style space $\mathcal{S}_b$. After merging them, LOMIT learns to reconstruct the original image. (b) For the cross-domain translation $\mathcal{X}_1 \rightarrow \mathcal{X}_2$, LOMIT combines a foreground style extracted from $\mathcal{X}_2$ with a content, background style code extracted from $\mathcal{X}_1$.

Furthermore, when multiple attributes are involved in an exemplar image, one has no choice but to impose all of them when translating a given image. For example, in a person's facial image translation, if the exemplar image has two attributes, (1) a smiling expression and (2) a blonde hair, then both attributes have to be transferred with no other options.

To tackle these issues, we propose a novel, LOcal Mask-based Image Translation approach, called LOMIT, which jointly generates a local, pixel-wise soft binary mask of an exemplar (i.e., the source region from which to extract out the style information) and that of an input image to translate (i.e., the target region to which to apply the extracted style). This approach has something in common with those recent approaches that have attempted to leverage an attention mask in image translation (Pumarola et al., 2018; Chen et al., 2018; Yang et al., 2018; Ma et al., 2018; Mejjati et al., 2018). In most approaches, the attention mask (extracted from an input image) plays a role of determining the target region to apply a translation. Note that we expand those approaches by jointly extracting two masks, from an input and an exemplar image, respectively, acting as the attention mask of an input and a relevant region (foreground) extractor of an exemplar.

The main novelty of LOMIT is that to jointly obtain the local masks of two images, we utilize the co-segmentation networks (Rother et al., 2006), which aim (1) to extract the targeted style information without noise introduced from irrelevant regions and (2) to translate only the necessary region of a target image while minimizing its distortion. While co-segmentation approaches were originally proposed to capture the regions of a common object existing in multiple input images (Rother et al., 2006; Li et al., 2018), we adopt and train co-segmentation networks for our own purpose.

Once obtained local masks, LOMIT extends a recently proposed technique for image translation, called adaptive instance normalization, using highway networks (Srivastava et al., 2015), which computes the weighted average of the input and the translated pixel values using the above-mentioned pixel-wise local mask values as different linear combination weights per pixel location. LOMIT has an additional advantage of being able to manipulate the computed masks to selectively transfer an intended style, e.g., choosing either a hair region (to transfer the hair color) or a facial region (to transfer the facial expression).

The effectiveness of LOMIT is evaluated on two facial datasets, via a user study and other quantitative measures such as the inception score and the classification accuracy.

## 2 BASIC SETUP

We define "content" as common features (an underlying structure) across all domains (e.g., the pose of a face, the location and the shape of eyes, a nose, a mouth, and hair), and "style" as a representation of the structure (e.g., background color, facial expression, skin tone, and hair color).

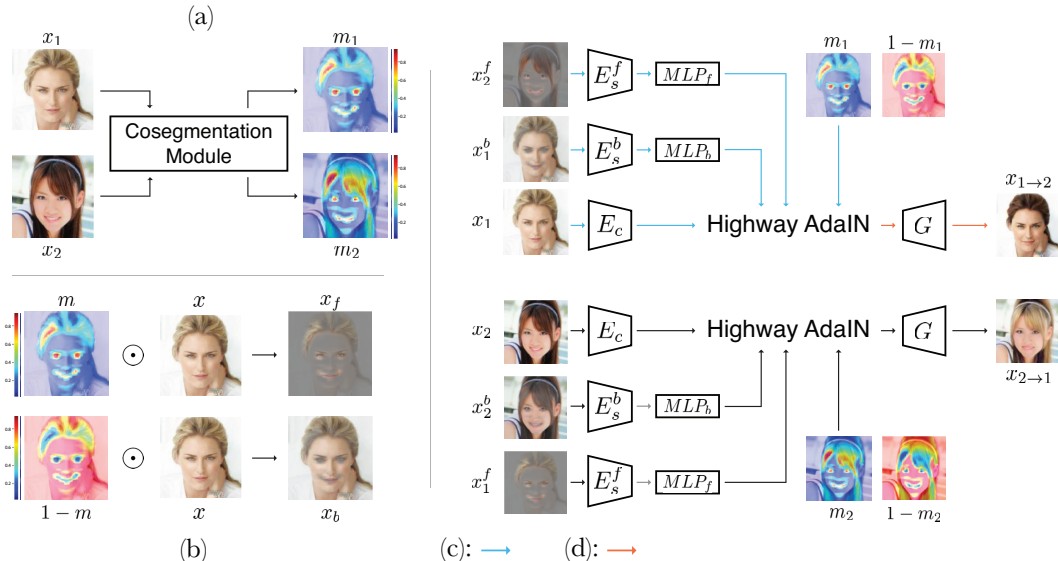

Figure 2: Image translation workflow. (a) First, LOMIT jointly generates masks for the input and the exemplar images through co-segmentation networks. (b) Next, we separate each image of $x_1$ and $x_2$ into the foreground and the background regions, depending on how much each pixel is involved in image translation. (c,d) By combining the content and the background style code from $x_1$ with the foreground style code $x_2$, we obtain a translated image $x_{1\to2}$. Note that LOMIT also learns the opposite-directional image translation $x_{2\to1}$ by interchanging $x_1$ and $x_2$. Finally, LOMIT learns image translation using the cycle consistency loss from $\mathcal{X}_1 \to \mathcal{X}_2 \to \mathcal{X}_1$ and $\mathcal{X}_2 \to \mathcal{X}_1 \to \mathcal{X}_2$.

As shown in Fig. 1, we assume that an image $x$ can be represented as $x = c \oplus s$, where $c$ is a content code in a content space, and $s$ is a style code in a style space. The operator $\oplus$ combines and converts the content code $c$ and the style code $s$ into a complete image $x$.

By considering the local mask indicating the relevant region (or simply, the foreground) to extract the style from or to apply it to, we further assume that $s$ is decomposed into $s = s_f \oplus s_b$, where $s_f$ is the style code extracted from the foreground region and $s_b$ is that from the background region. Separating an integrated style space $\mathcal{S}$ into a foreground style space $\mathcal{S}_f$ and a background style space $\mathcal{S}_b$ play a role of disentangling style feature representation[1].

The pixel-wise soft binary mask $m$ of an image $x$ is represented as a matrix with the same spatial resolution of $x$. Each entry of $m$ lies between 0 and 1, which indicates the degree of the corresponding pixel belonging to the foreground. Then, the local foreground/background regions $x^f/x^b$ of $x$ is obtained as

$$x^f = m \odot x, \quad x^b = (1 - m) \odot x, \tag{1}$$

where $\odot$ is an element-wise multiplication. Finally, our assumption is extended to $x = c \oplus s_f \oplus s_b$, where $c$, $s_f$, and $s_b$ are obtained by the content encoder $E_c$:, the foreground style encoder $E_s^f$, and the background style encoder $E_s^b$, respectively, which are all shared across multiple domains in LOMIT, i.e.,

$$\{c_x, s_x^f, s_x^b\} = \{E_c(x), E_s^f(x^f), E_s^b(x^b)\} \qquad c_x \in \mathcal{C}, s_x^f \in \mathcal{S}_f, s_x^b \in \mathcal{S}_b \tag{2}$$

It is critical in LOMIT to properly learn to generate the local mask involved in image translation. To this end, we propose to combine the mask generation networks with our novel highway adaptive instance normalization, as will be described in Section 3.2.

---

[1] To verify the representations are properly disentangled, we refer the readers to the 2D embedding visualization of each space $(\mathcal{C}, \mathcal{S}_f, \mathcal{S}_b)$ in Fig. 8 in in the appendix.

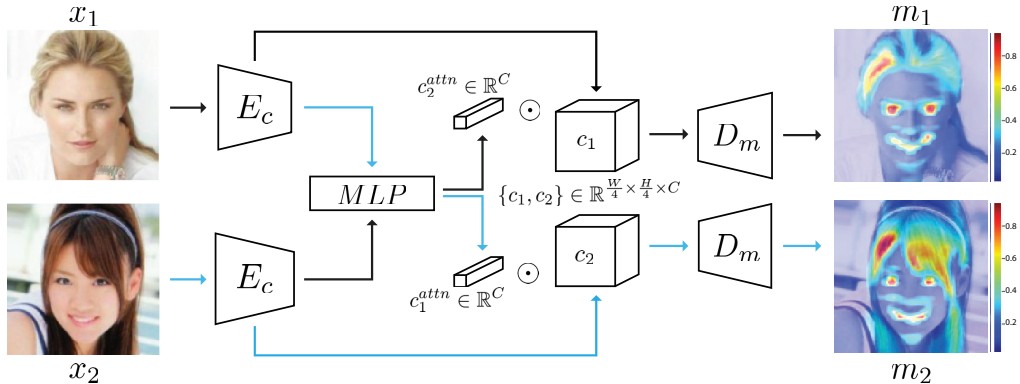

Figure 3: Local mask extraction via co-segmentation networks. The blue arrows indicate the forward propagation path in generating the mask $m_2$ of $x_2$, which relies on the global average pooled vector $c_1^{attn}$ of the content activation map $c_1$. The local masks of two images are jointly computed in an inter-dependent manner so that their style codes are interchangeable.

## 3 LOCAL IMAGE TRANSLATION MODEL

We first denote $x_1 \in \mathcal{X}_1$ and $x_2 \in \mathcal{X}_2$ as images from domains $\mathcal{X}_1$ and $\mathcal{X}_2$, respectively. As shown in Fig. 2, LOMIT converts a given image $x_1$ to $\mathcal{X}_2$ and vice versa, i.e., $x_{1 \to 2} = G(h(E_c(x_1), E_s^f(x_2^f), E_s^b(x_1^b)))$, and $x_{2 \to 1} = G(h(E_c(x_2), E_s^f(x_1^f), E_s^b(x_2^b)))$, where $G$ is decoder networks and $h$ is our proposed, local mask-based highway adaptive instance normalization (or in short, HAdaIN), as will be described in detail in Section 3.2.

For a brevity purpose, we omit the domain index notation in, say, $m = \{m_1, m_2\}$ and $x = \{x_1, x_2\}$, unless needed for clarification.

### 3.1 LOCAL MASK EXTRACTION

LOMIT utilizes the local mask $m$ to separate the image $x$ into the foreground and the background regions, $x^f$ and $x^b$. That is, we jointly extract the local masks of the input and the exemplar images, as those effectively involved in image translation, via co-segmentation networks. For example, given the input image and the exemplar, if LOMIT identifies the the hair color difference of a facial image, e.g., blonde vs. black, then, our local masks should be obtained as the hair regions from two images.

As shown in Fig. 3, given two images $x_1$ and $x_2$, the co-segmentation networks first encode the content of each as $c_1$ and $c_2$ via the content encoder $E_c$. Next, in the case of computing the segmentation of $x_2$, after average-pooling $c_1$ globally, we forward it to an MLP to obtain the channel-wise soft binary mask $c_1^{attn}$, which is then multiplied with $c_2$ in a channel-wise manner, i.e., $c_1^{attn} \odot c_2$, where $c_1^{attn} = \sigma(\text{MLP}(c_1))$. This step works as transferring the objects information from $x_1$ to $x_2$. Finally, we forward-propagate this output into the attention network $A$ to obtain the local mask $m_2$ of $x_2$, i.e., $m_2 = A(c_1^{attn} \odot c_2)$. The same process applies to the opposite case in a similar manner, resulting in $m_1 = A(c_2^{attn} \odot c_1)$.

Note that our co-segmentation networks are trained in an end-to-end manner with no direct supervision.

### 3.2 HIGHWAY ADAPTIVE INSTANCE NORMALIZATION

Adaptive instance normalization is an effective style transfer technique (Huang & Belongie, 2017). Generally, it matches the channel-wise statistics, e.g., the mean and the variance, of the activation map of an input image with those of a style image. In the context of image translation, MU-NIT (Huang et al., 2018) extends AdaIN in a way that the target mean and the variance are computed

as the outputs of the trainable functions $\beta$ and $\gamma$ of a given style code, i.e.,

$$\text{AdaIN}_{x_1 \to x_2}(c_1, s_2) = \gamma(s_2) \left( \frac{c_1 - \mu(c_1)}{\sigma(c_1)} \right) + \beta(s_2), \tag{3}$$

As we pointed out earlier, such a transformation is applied globally over the entire region of an image, which may unnecessarily distort irrelevant regions. Hence, we formulate our local mask-based highway AdaIN (HAdaIN) as

$$\text{HAdaIN}_{x_1 \to x_2}(m_1, c_1, s_2^f, s_1^b) = m_1 \odot \text{AdaIN}_{x_1 \to x_2}(c_1, s_2^f) + (1 - m_1) \odot \text{AdaIN}_{x_1 \to x_1}(c_1, s_1^b), \tag{4}$$

where each of $\beta$ and $\gamma$ in Eq. (3) is defined as a multi-layer perceptron (MLP), i.e., $[\beta(s_f); \gamma(s_f)] = \text{MLP}_f(s_f)$ and $[\beta(s_b); \gamma(s_b)] = \text{MLP}_b(s_b)$. Note that we use different MLPs for the foreground and the background style code inputs. The first term of Eq. (4) corresponds to the local region of an input image translated by the foreground style, while the second corresponds to the complementary region where the original style of the input is kept as it is.

## 4 TRAINING OBJECTIVES

This section describes each of our loss terms in the objective function used for training our model.

### 4.1 STYLE AND CONTENT RECONSTRUCTION LOSS

The foreground style of the translated output should be close to that of the exemplar, while the background style of the translated output should be close to that of the original input image. We formulate this criteria as the following style reconstruction loss terms:

$$\mathcal{L}_{s_f}^{1 \to 2} = \mathbb{E}_{x_{1 \to 2}^f, x_2^f} [\|E_s^f(x_{1 \to 2}^f) - E_s^f(x_2^f)\|_1] \tag{5}$$

$$\mathcal{L}_{s_b}^{1 \to 2} = \mathbb{E}_{x_{1 \to 2}^b, x_1^b} [\|E_s^b(x_{1 \to 2}^b) - E_s^b(x_1^b)\|_1]. \tag{6}$$

From the perspective of content information, the content feature of an input image should be consistent with its translated output, which is represented as the content reconstruction loss as

$$\mathcal{L}_c^{1 \to 2} = \mathbb{E}_{x_{1 \to 2}, x_1}[\|E_c(x_{1 \to 2}) - E_c(x_1)\|_1]. \tag{7}$$

Note that the content reconstruction is imposed across the entire region of the input image, regardless of the local mask.

### 4.2 IMAGE RECONSTRUCTION LOSS

As an effective supervision approach in an unpaired image translation setting, we adopt the image-level cyclic consistency loss (Zhu et al., 2017a) between an input image and its output through two consecutive image translations of $\mathcal{X}_1 \to \mathcal{X}_2 \to \mathcal{X}_1$ (or $\mathcal{X}_2 \to \mathcal{X}_1 \to \mathcal{X}_2$), i.e.,

$$\mathcal{L}_{cyc}^{1 \to 2 \to 1} = \mathbb{E}_{x_1} [\|x_{1 \to 2 \to 1} - x_1\|_1]. \tag{8}$$

Meanwhile, similar to previous studies (Huang et al., 2018; Lee et al., 2018), we translate not only $(x_1 \to x_{1 \to 2})$ but also $(x_1 \to x_{1 \to 1})$. This intra-domain translation $(x_1 \to x_{1 \to 1})$ should work similarly to auto-encoder (Larsen et al., 2016), and the corresponding loss term is written as

$$\mathcal{L}_x^{1 \to 1} = \mathbb{E}_{x_1} [\|x_{1 \to 1} - x_1\|_1] \tag{9}$$

### 4.3 DOMAIN ADVERSARIAL LOSS

To approximate the real-data distribution via our model, we adopt the domain adversarial loss by introducing the discriminator networks $D_{src}$. Among the loss terms proposed in the original GAN(Goodfellow et al., 2014), LSGAN(Mao et al., 2017), and WGAN-GP(Arjovsky et al., 2017; Gulrajani et al., 2017), we chose WGAN-GP, which is shown to empirically work best, as an adversarial method. That is, our adversarial loss is written as

$$\mathcal{L}_{adv}^{1 \to 2} = \mathbb{E}_{x_1} [D_{src}(x_1)] - \mathbb{E}_{x_{1 \to 2}} [D_{src}(x_{1 \to 2})] - \lambda_{gp} \mathbb{E}_{\hat{x}}[(\|\nabla_{\hat{x}} D_{src}(\hat{x})\|_2 - 1)^2], \tag{10}$$

where $x_{1 \to 2} = G(h(c_1, s_2^f, s_1^b))$, $\hat{x}$ is a sampled value from the uniform distribution, and $\lambda_{gp} = 10$. Also, we apply the loss proposed in patchGAN (Isola et al., 2017; Zhu et al., 2017a).

### 4.4 MULTI-ATTRIBUTE TRANSLATION LOSS

we use an auxiliary classifier (Odena et al., 2016) to cover multi-attribute translation with a single shared model, similar to StarGAN (Choi et al., 2018). The auxiliary classifier $D_{cls}$, which shares the parameters with the discriminator $D_{src}$ except for the last layer, classifies the domain of a given image. In detail, its loss term is defined as

$$\mathcal{L}_{cls_r}^{1\rightarrow2} = \mathbb{E}_{x_1}\left[-\log D_{cls}(y_{x_1}|x_1)\right] \tag{11}$$

$$\mathcal{L}_{cls_f}^{1\rightarrow2} = \mathbb{E}_{x_{1\rightarrow2}}\left[-\log D_{cls}(y_{x_2}|x_{1\rightarrow2})\right], \tag{12}$$

where $y_x$ is the domain label of an input image $x$. Similar to the concept of weakly supervised learning (Zhou et al., 2016; Selvaraju et al., 2017), This loss term plays a role of supervising the local mask $m$ to point out the proper region of the corresponding domain $y$ through the HAdaIN module, allowing our model to extract out the style from its proper region of the exemplar.

### 4.5 MASK REGULARIZATION LOSSES

We impose several additional regularization losses on local mask generation to improve the overall image generation performance as well as the interpretability of the generated mask.

The first regularization is to minimize the difference of the mask values of those pixels that have similar content information. This helps the local mask consistently capture a semantically meaningful region as a whole, e.g., capturing the entire hair region even when the lighting conditions and the hair color vary significantly within the exemplar. In detail, we design this regularization as minimizing

$$\mathcal{R}_1 = \mathbb{E}\left[\sum_{i=1,\cdots,W,j=1,\cdots,H}\left[|(m\cdot\vec{1}^T)-(\vec{1}\cdot m^T)|\odot(\hat{c}\cdot\hat{c}^T)\right]_{ij}\right] \tag{13}$$

where $\vec{1}$ is a vector whose elements are all ones, $\{\vec{1},m\}\in\mathbb{R}^{WH\times1}$, and $\hat{c}\in\mathbb{R}^{WH\times C}$ where $\hat{c}=\frac{c}{\|c\|}$. The first term is the distance matrix of all the pairs of pixel-wise mask values in $m$, and the second term is the cosine similarity matrix of all the pairs of $C$-dimensional pixel-wise content vectors. Note that we backpropagate the gradients generated by this regularization term only through $m$ to train the co-segmentation networks, but not through $\hat{c}$, which does not affect the encoder $E$.

The second regularization is to make the local masks of the two images capture only those regions having contrasting styles. This regularization is useful especially when multiple attributes are involved in image translation. For example, if the two facial images have different hair colors but common facial expressions, then the local mask should indicate only the hair region. We formulate this regularization by maximizing the style difference between the local mask of two images, which is written as

$$\mathcal{R}_2 = -\mathbb{E}\left[\|s_1^f - s_2^f\|_1\right] \tag{14}$$

The third regularization is simply to minimize the local mask region (Chen et al., 2018; Pumarola et al., 2018) to encourage the model to focus only on a necessary region involved in image translation, by minimizing

$$\mathcal{R}_3 = \mathbb{E}\|m\|_1 \tag{15}$$

### 4.6 FULL LOSS

Finally, our full loss is defined as

$$\mathcal{L}_D = -\mathcal{L}_{adv} + \lambda_{cls}\mathcal{L}_{cls_r},$$
$$\mathcal{L}_G = \mathcal{L}_{adv} + \lambda_{cls}\mathcal{L}_{cls_f} + \lambda_{s,c}(\mathcal{L}_{s_f} + \mathcal{L}_{s_b} + \mathcal{L}_c) + \lambda_x(\mathcal{L}_{cyc} + \mathcal{L}_x^{1\rightarrow1} + \mathcal{L}_x^{2\rightarrow2}) \tag{16}$$
$$+\lambda_1\mathcal{R}_1 + \lambda_2\mathcal{R}_2 + \lambda_3\mathcal{R}_3,$$

where $\mathcal{L}$ without superscript denotes $(1\rightarrow2,2\rightarrow1)$, $\lambda_{cls}=1$, $\lambda_{s,c}=1$, $\lambda_x=10$, $\lambda_1=0.1$, $\lambda_2=0.01$, and $\lambda_3=0.0001$. Note that our training process contains both intra-domain translation, $(x_1\rightarrow x_{1\rightarrow1}$ and $x_2\rightarrow x_{2\rightarrow2})$, and inter-domain translation, $(x_1\rightarrow x_{1\rightarrow2}$ and $x_2\rightarrow x_{2\rightarrow1})$.

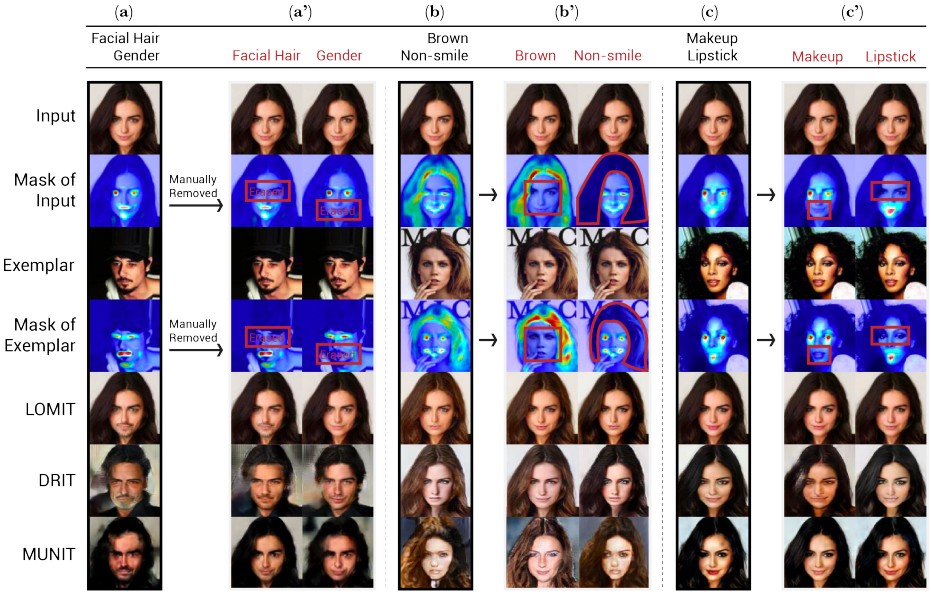

Figure 4: Comparison with the baselines on CelebA (Liu et al., 2015) dataset. Each column denotes the input and output image of LOMIT and other baseline models, corresponding with the target attribute shown in the top. Meanwhile, each row indicates the input and output image followed by each of masks and the generated image of LOMIT and the baseline models.

## 5 EXPERIMENTS

We conduct LOMIT and other baseline models on two facial datasets, CelebA (Liu et al., 2015) and EmotioNet (Fabian Benitez-Quiroz et al., 2016). We first describe the datasets and the baseline models in subsection 5.1, 5.2. Second, we present the qualitative comparisons of both multi- and single-attribute translation results against baseline methods in subsection 5.3. Third, we report the user study results to validate the human-perceived quality of the translated results in subsection 5.4. Lastly, we evaluate the performances of LOMIT using the inception score (Salimans et al., 2016) and the classification accuracy. The model architecture and training details we set in the experiments are described in appendix (subsection 7.1, 7.2).

### 5.1 DATASETS

**CelebA.** The CelebA (Liu et al., 2015) dataset consists of 202,599 face images of celebrities and 40 attribute annotations per image. We pick 10 attributes (i.e., black_hair, blond_hair, brown_hair, smiling, goatee, mustache, no_beard, male, heavy_makeup, wearing_lipstick) that would convey meaningful local masks. We randomly select 2,000 images for testing and use the others for training. Images are center-cropped and scaled down to 128×128.

**EmotioNet.** The EmotioNet (Fabian Benitez-Quiroz et al., 2016) dataset contains 975,000 images of facial expressions in the wild, each annotated with 12 action units (AUs). Each AU denotes an activation of a specific facial muscle (e.g., jaw drop, nose wrinkler). We crop each image using a face detector [2] and resize them to 128×128. We use 2,000 images for testing and 200,000 images for training.

### 5.2 BASELINE METHODS

**MUNIT.** MUNIT (Huang et al., 2018) decomposes an image into the domain-invariant content code and the domain-specific style code. Involving random sampling for latent style codes while

---

[2]https://github.com/ageitgey/face_recognition

Expressionless (1,4,25) $\longrightarrow$ Happy (6,12,25)      Happy (6,12,25) $\longrightarrow$ Expressionless (1,2,4,25,26)

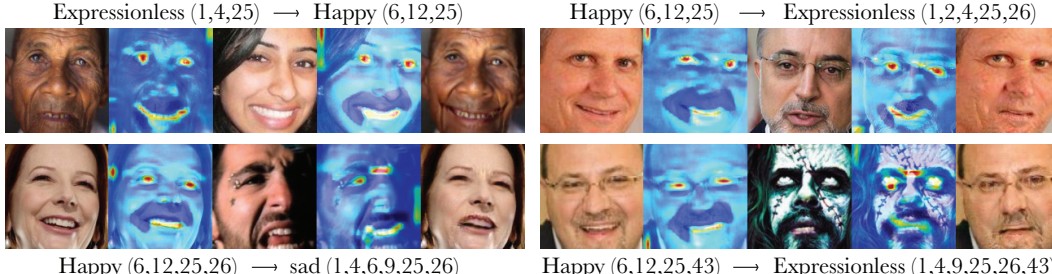

Happy (6,12,25,26) $\longrightarrow$ sad (1,4,6,9,25,26)      Happy (6,12,25,43) $\longrightarrow$ Expressionless (1,4,9,25,26,43)

Figure 5: The result of action unit translation using EmotioNet dataset (Fabian Benitez-Quiroz et al., 2016). In each part, the first, the third, and the last images are an input image, an exemplar image, and a translated output, respectively. Each number in parenthesis indicates AU.

training, MUNIT attempts to reflect the multimodal nature of various style domains. We implement MUNIT to be trained on CelebA (Liu et al., 2015) dataset and report results for our comparison.

**DRIT.** DRIT (Lee et al., 2018) employs two encoders of which each extracts the domain-invariant content information and domain-specific style information, respectively. The model is trained leveraging a content discriminator that ensures the content space to be shared. Loss functions and training processes are similar to MUNIT.

### 5.3 QUALITATIVE RESULTS

**CelebA.** As shown in Fig. 4, we compare our model with the baseline models using CelebA dataset (Liu et al., 2015). The baseline models are trained with a dataset corresponding to each target attribute at the topmost column (e.g., in the gender case, a dataset is divided into male/female). On the other hand, when training LOMIT, we construct a set of multiple attributes by combining a few different attributes and train the model for multi-attribute translation (i.e., the columns in a black border, (a), (b), and (c) in Fig. 4). Meanwhile, in order to conduct the single-attribute translation, we interactively modify the output masks of the co-segmentation module and forward the manipulated masks into the networks. That is, as illustrated in Figs. 4(a'), (b'), and (c'), we manually remove the mask for both an input and an exemplar and obtain the result for single-attribute translation. Note that the area marked as a red rectangle in the mask indicates the removed area.

The first four rows correspond to the input images, their generated masks, the exemplars, and their generated masks. Each of the last three rows provides comparisons of our model against the baselines. The topmost row indicates the target attribute for each column. Those in black denote multiple attributes while those in red represent a single attribute after removing the mask subregion (a red rectangle) corresponding to the other attribute. We denote Facial Hair when belonging to any of the three classes, Beard, Goatee or Mustache. LOMIT tends to keep the background intact across various classes and apply the style to the appropriate region, while transferring the properly extracted style (attribute) from the exemplar. When compared to LOMIT, the two other models suffer from undesirable distortion in the background as shown in the first and the second rows from the bottom of (a), (a'). Meanwhile, as can be seen in the bottommost row of (e), MUNIT fails to apply the style to the suitable region due to the lack of an attention mask (through the highway networks). The images of DRIT in the columns (e') show a translation through the improperly extracted style because the hair region on the images contains a white color which seems to be referenced from the shoulder of a person in the exemplar. It indicates that a mask for the exemplar should be properly incorporated in the process. From the comparison with the baseline models, we justify the needs of the local masks and the HAdaIN module of LOMIT.

**EmotioNet.** Fig. 5 shows the results for AU translation. For the training, we use each AU (1, 2, 4, 5, 6, 9, 12, 17, 20, 25, 26, and 43) as a label for the multi-attribute translation loss, so that the model can be trained for translating multi-AUs from the exemplar. Each section is composed of an input image, its mask, an exemplar, its mask, and a translated output. For example, the left top part, the input containing AUs 1, 4, 25 (Expressionless) takes the exemplar whose AUs are 6, 12, 25 (Happy).

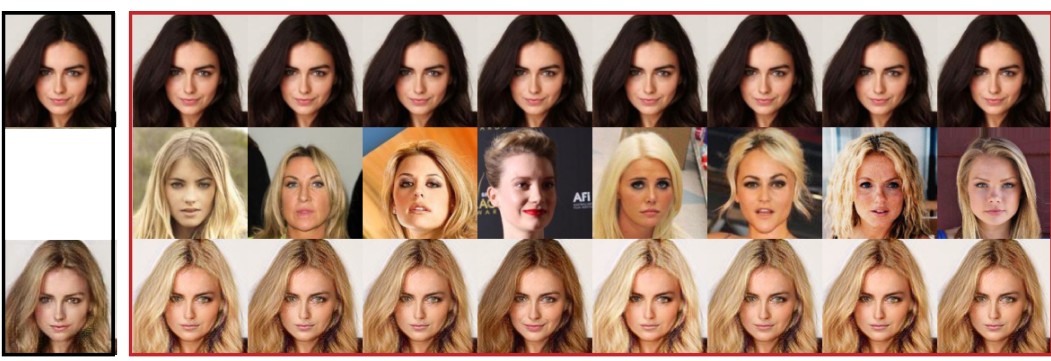

StarGAN                                                    LOMIT

Figure 6: Comparison with StarGAN. The results demonstrates that LOMIT achieves the multi-modality while maintaining a high-quality output. On the other hand, StarGAN (Choi et al., 2018) is only able to generate an unimodal output.

The translated output demonstrates that it preserves the identity of the input image while properly transferring the AUs of the exemplar. From the results, we verify LOMIT can be generally used in facial attribute translation. Note that we do not modify the masks in AUs translation. The dark patterns on the mouth corresponding to AU 20 have been generated because the region on the mouth was not involved in the translation during training.

**Comparison with StarGAN.** Because StarGAN (Choi et al., 2018) is a state-of-the-art method in attribute translation, we qualitatively compare it with LOMIT to verify the superiority of our method. In Fig. 6, each row from the top indicates an input, an exemplar, and a translated output. The leftmost column surrounded by the black rectangle denotes the result of StarGAN while the rest of columns in the red rectangle correspond to our results. StarGAN is only able to generate an unimodal output depending on an input multi-hot vector indicating a target attribute. On the other hand, LOMIT generates diverse outputs reflecting each corresponding exemplar. As can be seen in the figure, the hair color between each of outputs and its corresponding exemplar is considerably similar showing the superiority of LOMIT. We believe the multimodal translation can be achieved by the training objectives of LOMIT. First, the adversarial loss and the multi-attribute translation loss encourage the model to generate a blonde person because the former reduces the distance between the distribution of the generated image and that of a real image containing a blonde hair. On the other hand, the latter makes the model generate an image with a blonde hair to be classified as being blonde. Second, the image reconstruction loss and the style reconstruction loss encourage the model to keep an intrinsic style of the exemplar. Specifically, the image reconstruction loss forces a reconstructed image to contain the same pixel value with an input image. Meanwhile, the style reconstruction loss makes a style code of the exemplar be kept after being applied to the input image. That is, each style code of different hair color has to be maintained to minimize the loss. These aspects allow LOMIT to suitably learn how to translate an image while achieving the multimodality. Note that LOMIT is able to cover the intra-domain variation even when an unseen style is given in an exemplar because each exemplar in Fig 6 is sampled from the test dataset.

## 5.4 QUANTITATIVE EVALUATION

**User Study.** To evaluate the effectiveness of the proposed method, we conduct two AB tests by comparing LOMIT with other baseline models. First, we randomly sample a hundred pairs composed of two images whose attributes are (Brown, Non-smile) and (Blonde, Smile), respectively from the test dataset. We then construct a database containing the translated results from (Brown, Non-smile) to (Blonde, Smile) by using the pairs as input to each model.

In the first test, we evaluate how realistic the translated images of each model are. We randomly sample 30 real images in (Blonde, Smile) from the entire dataset and 10 of the translated images per model from the database. Each time, participants are shown a pair of images, composed of the sampled real image and the translated image, between which to choose one that looks more realistic. The superior realism rate of LOMIT is reported in Fig. 7(a), which indicates that the translated results based on LOMIT look more natural in human eyes.

**(a)**

| Model | Realism Rate (%) |
|-------|------------------|
| DRIT  | 12.3 |
| MUNIT | 11.5 |
| LOMIT | 25.6 |

**(b)**

■ LOMIT  ■ DRIT  ■ MUNIT

**Q1 Which one has better applied the style to the image?** (%)

| 67.7 | 32.3 |
| 66.3 | 33.7 |

**Q2. Which one has better kept the background unchanged?**

| 95.7 | 4.3 |
| 96.7 | 3.2 |

Figure 7: User study results. Details are described in subsection. 5.4.

| Class | DRIT | | | MUNIT | | | LOMIT | | |
|-------|------|------|------|-------|------|------|-------|------|------|
| | IS | | CA | IS | | CA | IS | | CA |
| | Mean | Std | (%) | Mean | Std | (%) | Mean | Std | (%) |
| Facial Hair | **0.3295** | 0.24 | 23.8 | 0.2808 | 0.25 | 60.0 | 0.3105 | 0.27 | **71.4** |
| Gender | **0.2703** | 0.21 | 21.1 | 0.1368 | 0.19 | 53.0 | 0.2348 | 0.22 | **83.9** |
| Wearing Lipstick | **0.2685** | 0.20 | 19.9 | 0.1751 | 0.20 | 57.1 | 0.2528 | 0.20 | **73.7** |
| Facial Hair + Gender | **0.3805** | 0.24 | 14.3 | 0.1972 | 0.25 | 34.9 | 0.2069 | 0.23 | **68.1** |
| Makeup + Wearing Lipstick | **0.2853** | 0.21 | 16.1 | 0.2472 | 0.24 | 27.3 | 0.2834 | 0.22 | **72.8** |

Table 1: Comparisons for Inception Score (IS) and Classification Accuracy (CA)

For the second test, we randomly sample ten pairs of the translated images from two different models (e.g., LOMIT / DRIT or LOMIT / MUNIT) with corresponding inputs. Given specific questions such as Q1 or Q2 as shown in Fig. 7 (b), users confirm that LOMIT does not only apply exemplar styles better, but also maintains the background region unaffected. Both (a) and (b) verify the effectiveness of the algorithms of LOMIT, such as HAdaIN and the disentangled style encoders, because we extract the relevant style and apply it to the relevant area (foreground) of the content code while maintaining the rest.

**Inception score and classification accuracy.** We compare LOMIT with the baselines using inception score (Salimans et al., 2016) (IS) and classification accuracy (CA). IS is high if translated images are diverse and high quality. We follow the procedure in MUNIT (Huang et al., 2018) to obtain IS. For the classification, we use the pretrained Inception-v3 (Szegedy et al., 2016) and fine-tune with CelebA (Liu et al., 2015) dataset. To be classified well with high accuracy, a translated image must have appropriate attribute in the exemplar. Table 1 lists up the resulting scores and accuracies. In terms of the CA, LOMIT achieves the highest accuracy across all classes evaluated by large margins. DRIT achieves slightly higher IS than LOMIT, but in the cost of the CA. It indicates that DRIT produces diverse outputs, however with less recognizable image outputs.

## 6  CONCLUSIONS

In this work, we proposed a local mask-based image-to-image translation model called LOMIT. The co-segmentation networks jointly generate the mask of an input image and that of an exemplar, respectively. A mask of the exemplar exclude an irrelevant region to extract the style information from a relevant region. The other mask of an input captures the region to apply the style of an exemplar while maintaining an original style in the rest (through our highway adaptive instance normalization). LOMIT achieves outstanding results compared with the state-of-the-art methods (Huang et al., 2018; Lee et al., 2018). As future work, we will extend our approach as a general normalization method which can be used in other computer vision tasks.

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

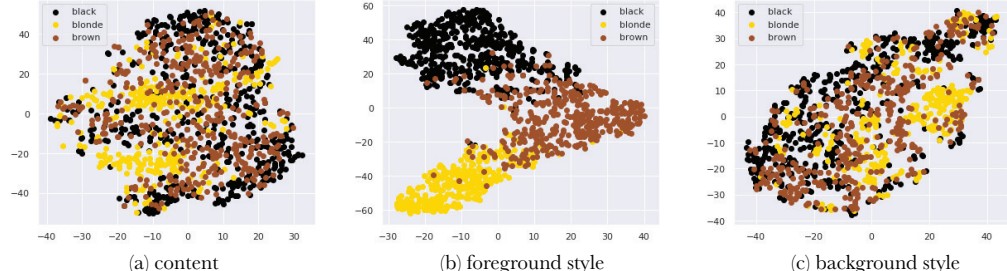

(a) content        (b) foreground style        (c) background style

Figure 8: t-SNE visualization of the content space $\mathcal{C}$ and foreground, background style space $\mathcal{S}_f$,$\mathcal{S}_b$. Only the foreground style space $\mathcal{S}_f$ unfolds the disentangled representations depending on the domain information. On the other hand, the content codes in $\mathcal{C}$ and the style codes in $\mathcal{S}_b$, extracted from an irrelevant region to the domain, show the domain-invariant representation.

# 7 APPENDIX

## 7.1 MODEL ARCHITECTURES

**Content Encoder.** Similar to MUNIT (Huang et al., 2018), the content encoder $E_c$ is composed of two strided-convolutional layers and four residual blocks (He et al., 2016). Following the previous approaches (Huang & Belongie, 2017; Nam & Kim, 2018), IN is used across all the layers in the content encoder.

**Style Encoders.** The style encoders $E_s^f, E_s^b$ have the same architecture but with different parameters. They consist of four strided-convolutional layers, a global average pooling layer, and a fully-connected layer. The style codes $s_f, s_b$ are eight-dimensional vectors. Also, style encoders $E_s^f, E_s^b$ share first few layers as the first few layers detect low-level feature. To maintain the style information, we do not use IN in the style encoders.

**Co-segmentation Networks.** Co-segmentation networks are composed of six convolutional layers with a batch normalization (Ioffe & Szegedy, 2015). MLP in Fig. 3 has two linear layers with tanh and sigmoid activation functions, respectively.

**Decoder.** Decoder $G$ has four residual blocks and two convolutional layers with an upsampling layer each. Because the layer normalization (LN) (Ba et al., 2016) normalizes the entire feature map, maintaining the differences between the channels, we use LN in the residual blocks for stable training.

**Discriminator.** Following StarGAN (Choi et al., 2018), the discriminator $D$ is composed of six strided-convolutional layers, followed by the standard discriminator and the auxiliary classifier.

## 7.2 TRAINING DETAILS

We utilize the Adam optimizer (Kingma & Ba, 2015) with $\beta_1 = 0.5$ and $\beta_2 = 0.999$. Following the state-of-the-art approach (Choi et al., 2018) in multi-attribute translation, we load the data with a horizontal flip with 0.5 percent. For stable training, we update $\{E_c, E_s^f, E_s^b, G\}$ in every five updates of $D$ (Gulrajani et al., 2017). We initialize the weights of $D$ from a normal distribution and apply the initialization (He et al., 2015) on others. Also, we use a batch size of eight and the learning rate of 0.0001. We linearly decay the learning rate by half in every 10,000 iterations from 100,000 iterations. All the models used in the experiments are trained for 200,000 iterations using a single NVIDIA TITAN Xp GPU for 30 hours each.

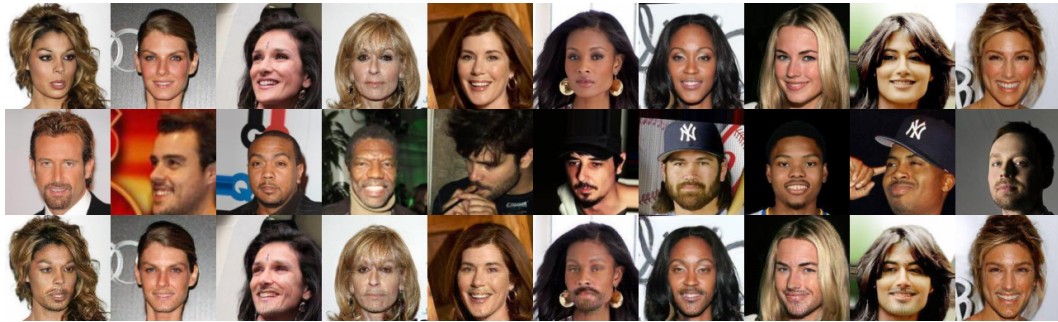

Figure 9: Additional result. The first row is the target image, the second row is the exemplar, and the Third row is a translated result. Note that the translated attributes are facial hair and gender.

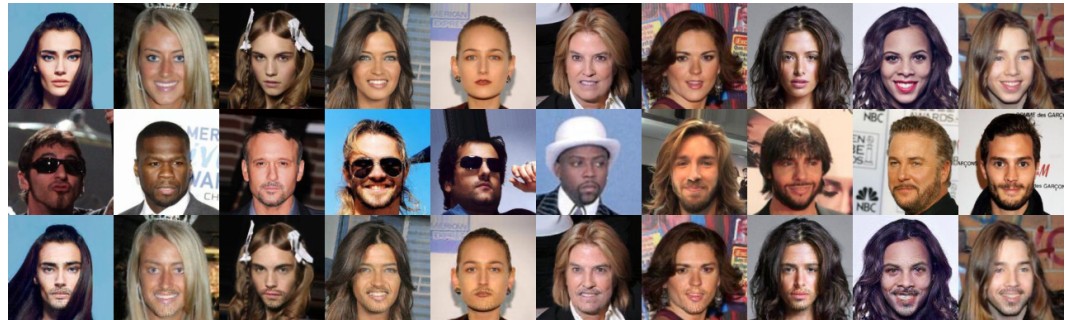

Figure 10: Additional result. The first row is the target image, the second row is the exemplar, and the Third row is a translated result. Note that the translated attributes are facial hair and gender.

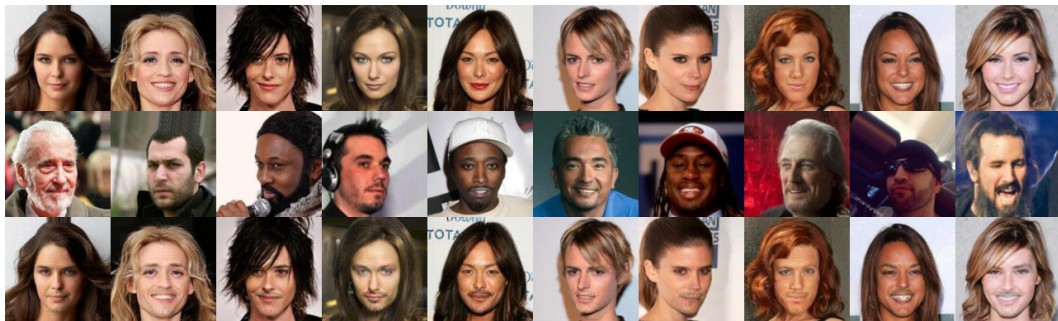

Figure 11: Additional result. The first row is the target image, the second row is the exemplar, and the Third row is a translated result. Note that the translated attributes are facial hair and gender.

