# OpenReview forum: "Local Image-to-Image Translation via Pixel-wise Highway Adaptive Instance Normalization"
_ICLR.cc/2019/Conference_

### Official Review · AnonReviewer1 · 2018-10-26
**Comparison and experiment setting are not explained well**

**Rating:** 5
**Confidence:** 5

**Review:**

This paper proposes an unpaired image-to-image translation method which applies the co-segmentation network and adaptive instance normalization techniques to enable the manipulation on the local regions.

Pros:
* This paper proposes to jointly learn the local mask to make the translation focus on the foreground instead of the whole image.
* The local mask-based highway adaptive instance normalization apply the style information to the local region correctly.

Cons:
* There seems a conflict in the introduction (page 1): the authors clarify that “previous methods [1,2,3] have a drawback of ....” and then clarify that “[1,2,3] have taken a user-selected exemplar image as additional input ...”.
* As the main experiments are about facial attributes translation, I strongly recommend to the author to compare their work with StarGAN [4].
* It is mentioned in the introduction (page 2) that “This approach has something in common with those recent approaches that have attempted to leverage an attention mask in image translation”. However, the differences between the proposed method with these prior works are not compared or mentioned. Some of these works also applied the mask technique or adaptive instance normalization to the image-to-image translation problem. I wonder the advantages of the proposed method compared to these works.
* The experiment setting is not clear enough. If I understand correctly, the face images are divided into two groups based on their attributes (e.g. smile vs no smile). If so, what role does the exemplar image play here? Since the attribute information has been modeled by the network parameters, will different exemplar image lead to different translation outputs?
* The github link for code should not provide any author information.

[1] Multimodal Unsupervised Image-to-Image Translation
[2] Diverse Image-to-Image Translation via Disentangled Representations
[3] Exemplar Guided Unsupervised Image-to-Image Translation
[4] StarGAN: Unified Generative Adversarial Networks for Multi-Domain Image-to-Image Translation

Overall, I think the proposed method is well-designed but the comparison and experiment setting are not explained well. My initial rating is weakly reject.

---

> ### Author Response · Authors · 2018-11-27
> **Point-by-point response addressing the issues**
>
> 1)
> ‘Issue of conflict in introduction'
> We understand this issue, so we have improved the part as follows:
> “Previous studies have achieved such multimodal outputs by adding a random noise (BicycleGAN, Zhu et al. [1]) or taking a user-selected exemplar image (MakeupGAN, Chang et al. [2]). Recently, MUNIT (Huang et al. [3]) and DRIT (Lee et al. [4]) combine those two approaches …”
>
> 2)
> ‘Topic of baseline (StarGAN)'
> The reason we excluded StarGAN (Choi et al. [5]) from a baseline is because it does not support multimodal outputs. However, as the reviewer advised, StarGAN is a state-of-the-art method in attribute translation, so we have conducted an additional experiment and updated the results in the paper. Please refer to Subsection 5.3 and Fig. 6 in the paper.
>
> 3)
> ‘Discussion on local mask'
> Compared to other approaches utilizing an attention mask [6, 7, 8, 9, 10], LOMIT jointly generates another attention mask for an exemplar. It plays a role of determining a relevant region from which to extract out a style. In order to justify the effectiveness of separating the style into a foreground and a background style, we have conducted an ablation test between LOMIT (equipped with a mask for an exemplar) and LOMIT_single (without the mask for an exemplar).
> First, we train each model for translating multi-attributes (Facial Hair and Gender). We then conduct the test using the inception score, the results are shown as follows.
>
>  	                 	        Facial Hair       	        Gender        	            FH+G
>  	LOMIT     	   :   (0.3105, 0.2697)  |  (0.2348, 0.2173)  |  (0.2069, 0.2323)
>  	LOMIT_single   :   (0.3040, 0.2556)  |  (0.2260, 0.2150)  |  (0.2029, 0.2343)
>
> , where the numbers in parentheses denote the mean and the standard deviation respectively. As can be seen, LOMIT shows the better results compared with LOMIT_single in both single- and multi-attribute translation. Based on these results, we verify that utilizing the mask of an exemplar improves the performance of our model.
>
> 4)
> ‘Diverse, different outputs due to exemplar images within a single domain'
> The role of an exemplar image is to convey the variation information within a particular attribute (bright vs. relatively dark in the case of a blonde hair), which allows to generate multiple possible translation outputs within a single attribute. This is so-called multi-modality in image translation, which our method as well as other existing methods, such as DRIT and MUNIT, has in common. Specifically, these models including outs decompose (or disentangle) the image into a content and a style features. Due to its reconstruction loss not only between x_1 and x_{1->2->1} but also between x_2 and x_{2->1->2}, the style feature (say, in x_2) should convey not only the domain label information but also the details within its domain.  We clarified such discussion in Subsection 5.3.
>
> 5)
> ‘Issue of github page'
> We sincerely apologize for the mishap, and we have changed the URL to an anonymous one.
>
>
> We have uploaded an updated paper containing several improvements which we have highlighted.
> Finally, the demo website of LOMIT can be found at http://123.108.168.4:5000
>
> [1] Toward multimodal image-to-image translation.
> [2] Pairedcyclegan: Asymmetric style transfer for applying and removing makeup.
> [3] Multimodal unsupervised image-to-image translation.
> [4] Diverse image-to-image translation via disentangled representations.
> [5] Stargan: Unified generative adversarial networks for multi-domain image-to-image translation.
> [6] Ganimation: Anatomically-aware facial animation from a single image.
> [7] Attention-gan for object transfiguration in wild images.
> [8] Unsupervised image translation with self-regularization and attention.
> [9] Exemplar guided unsupervised image-to-image translation.
> [10] Unsupervised attention-guided image to image translation.

---

> > ### Comment · AnonReviewer1 · 2018-11-30
> > **Rating unchanged**
> >
> > Thanks for your response and the revision. Some issues are fixed but the results are still not convincing. As to the comparison with StarGAN in Fig. 6, the output hair color is not so consistent to the exemplar image in some cases,  and the diversity in the blonde attribute is naturally very limited. Considering that one core contribution of this paper is the diversity controlled by the exemplar image, the paper should show how the output will vary given different exemplar images in the core experiments.
> > In addition, the inception score improvement from LOMIT_single to LOMIT is also limited, especially for the multi-attribute translation (FH+G) setting.
> > Therefore, I keep my initial rating.

---

> > > ### Author Response · Authors · 2018-12-19
> > > **A response addressing the issues**
> > >
> > > ‘Issue of multimodality depending on given exemplars’
> > > To address this issue, we improved Fig. 6, as can be found at http://123.108.168.4:5000/figure/page/6
> > > In The left macro column of the figure, (a) indicates the hair color translation result from brown to blonde while (b) represents the translation from non-facial hair to facial hair. LOMIT shows an outstanding performance compared to the baseline models in both reflecting the style of an exemplar and keeping the irrelevant region, such as a background and a face in the hair color translation, intact. Specifically, the third and the fifth columns of (a) and the second and the fifth columns of (b) show the results due to the noise in the style information extracted from the background. Besides, they also apply the extracted style to the irrelevant region of the input images, distorting the color and the tone of the face. On the other hand, the results of the baseline models in (b) show the inconsistent appearances of the facial hair with the exemplars, and less diversity in the results, compared to LOMIT.
> > >
> > > ‘Topic of inception score’
> > > When using multiple attributes, the associated region generally tends to be large. For example, transferring not just a facial hair but also a gender attribute would involve almost all the face region as the generated local mask. Thus, the idea of using only the partial region of an image involved in image translation may not have much impact on the performance improvement in the case of multi-attribute translation, compared to a single attribute translation. Nonetheless, the ablation test demonstrates a better performance though not outstanding. Furthermore, through a mask used for an exemplar, LOMIT enables the user to choose a style to transfer from the exemplar, which can be found at the rightmost column of the first and the second macro columns in Fig. 4 (please refer to http://123.108.168.4:5000/figure/page/4). We believe this approach has great potentials in diverse applications. For example, the technique can be effectively applied when the different styles in the same attribute (e.g., brown and blonde hair colors) co-exist in an exemplar. By explicitly specifying a style to transfer, a user can designate a concrete target style, and the model can conduct an appropriate translation in terms of the user.
> > >
> > > [1] Multimodal unsupervised image-to-image translation.
> > > [2] Diverse image-to-image translation via disentangled representations.

---

### Official Review · AnonReviewer3 · 2018-10-31
**paper should be improved before publishing**

**Rating:** 5
**Confidence:** 4

**Review:**

Summary--
The paper tries to address an issue existing in current image-to-image translation at the point that different regions of the image should be treated differently. In other word, background should not be transferred while only foreground of interest should be transferred. The paper propose to use co-segmentation to find the common areas to for image translation. It reports the proposed method works through experiments.

There are several major concerns to be addressed before considering to publish.

1) The paper says that "For example, in a person’s facial image translation, if the exemplar image has two attributes, (1) a smiling expression and (2) a blonde hair, then both attributes have to be transferred with no other options", but the model in the paper seems still incapable of transferring only one attribute. Perhaps an interactive transfer make more sense, while co-segmentation does not distinguish the part of interest to the user. Or training a semantic segmentation make more sense as the semantic segment can specify which region to transfer.

2) As co-segmentation is proposed to "capture the regions of a common object existing in multiple input images", why does the co-segmentation network only capture the eye and mouth part in Figure 2 and 3, why does it capture the mouth of different shape and style in the third macro column in Figure 4 instead of eyes? How to train the co-segmentation module, what is the objective function? Why not using a semantic segmentation model?

3) The "domain-invariant content code" and the "style code" seem rather subjective. Are there any principles to design content and style codes? In the experiments, it seems the paper considers five styles to transfer as shown in Table 1. Is the model easy to extend to novel styles for image translation?

4) What does the pink color mean in the very bottom-left or top-right heatmap images in Figure 2? There is no pink color reference in the colorbar.

5) Figure 5: Why there is similariy dark patterns on the mouth? Is it some manual manipulation for interactive transfer?

6) Though it is always good to see the authors are willing to release code and models, it appears uncomfortable that github page noted in the abstract reveals the author information. Moreover, in the github page,
even though it says "an example is example.ipynb", the only ipynb file contains nothing informative and this makes reviewers feel cheated.

Minor--
There are several typos, e.g., lightinig.

---

> ### Author Response · Authors · 2018-11-27
> **Point-by-point response addressing the issues**
>
> 1)
> ‘The capability of sigle-attribute translation '
> If LOMIT is trained for a multi-attribute translation, it does not normally support a single-attribute translation. However, as discussed in the introduction as well as pointed out by reviewer, LOMIT allows a user to perform this task by manually editing the mask. For example, to transfer only the facial expression but not the hair color, one can remove the hair region in the mask of an exemplar while keeping the mouth and eye regions. The details are shown in Fig. 4 as well as in Subsection 5.3.
>
> ‘Issue of semantic segmentation'
> The proposed idea of adopting the semantic segmentation is reasonable because it may generate different masks for different attributes. However, to perform it, pre-defined labels are explicitly required while LOMIT learns to extract the masks for both an input and an exemplar images without any direct supervison on the masks. In this sense, the co-segmentation module we proposed in LOMIT has more flexibility and extensibility in extracting masks corresponding to diverse domains than semantic segmentation.
>
> 2)
> ‘How to train co-segmentation module'
> The co-segmentation module is trained in a completely end-to-end manner, without any direct supervision. Similar to the previous studies (Pumarola et al. [1], Chen et al. [2], Yang et al. [3], Ma et al. [4], Mejjati et al. [5]), how it works is related to the domain adversarial loss and the multi-attribute translation loss. Suppose we are translating the hair color from black to blonde. Both losses encourage the model to generate a blonde person by minimizing each loss. However, LOMIT performs translation only to the target region of a mask (extracted by the co-segmentation module) through the highway adaptive instance normalization. Thus, in order to minimize the losses, the networks learn to generate a proper region as a mask.
>
> ‘Explanation on capturing the eyes and mouth in Fig. 2 and Fig. 3'
> The masks in Fig. 2 and Fig. 3 only capture the mouth, eyes and hair because the used attribute label during training consists of `Smile’ and `Hair color’.
>
> ‘Issue of capturing different shape in Fig. 4'
> The input of the co-segmentation module is a content code, so the region of mouth from two other content codes does not contain a different shape. Note that a content encoder of LOMIT is trained for encoding a common underlying structure except a style, such as the mustache.
>
> 'Possibility of using semantic segmentation'
> Due to the reasons discussed above, we do not think that the semantic segmentation can be an alternative to the proposed co-segmentation module in LOMIT.
>
> 3)
> ‘Issue of disentangled representations'
> Following MUNIT (Huang et al. [6]) and DRIT (Lee et al. [7]), we decompose an input image into the content and style codes. We define “content” as common underlying structure across all domains (e.g., pose of a face, location and shape of eyes), and “style” as a representation of the structure (e.g., color and facial expression). We have added the definitions of each content and style. Please refer to Section 2.
>
> ‘Discussion on extendibility'
> As other models (MUNIT [6], DRIT [7], StarGAN (choi et al. [8])), LOMIT cannot perform translation involving unseen labels. However, our model can cover an intra-domain variation though an unseen style is taken from an exemplar. Fig. 6 we have updated can clarify the point.
>
> 4)
> ‘Topic of pink color in heatmap image'
> The pink color in the figure comes from the background color of an image. Specifically, to visualize the heatmap image, we overlaid a mask on a corresponding image. Thus, the background color (pink) of the image affected the heatmap image. To alleviate the problem, we have replaced the image of the figures. Please refer to Fig. 2 and Fig. 3.
>
> 5)
> ‘Explanation on dark pattern'
> The dark patterns on the mouth corresponding to AU 20 have been generated because the region on the mouth was not involved in the translation during training. To clarify the points, we have updated Fig. 5 and its description in Subsection 5.3.
>
> 6)
> ‘Issue of github page'
> We sincerely apologize for the mishap, and we have changed the URL to an anonymous one. Furthermore, we supplement readme and ipynb file to avoid any uncertainty.
>
>
> We have uploaded an updated paper containing several improvements which we have highlighted.
> Finally, the demo website of LOMIT can be found at http://123.108.168.4:5000
>
> [1] Ganimation: Anatomically-aware facial animation from a single image.
> [2] Attention-gan for object transfiguration in wild images.
> [3] Unsupervised image translation with self-regularization and attention.
> [4] Exemplar guided unsupervised image-to-image translation.
> [5] Unsupervised attention-guided image to image translation.
> [6] Multimodal unsupervised image-to-image translation.
> [7] Diverse image-to-image translation via disentangled representations.
> [8] Stargan: Unified generative adversarial networks for multi-domain image-to-image translation.

---

> > ### Comment · AnonReviewer3 · 2018-12-01
> > **rating updated, more improvement needed**
> >
> > Thanks for your response and the revision. My rating is updated. However, here are further confusions/questions regarding to the rebuttal answers. These may be helpful for improving the paper further.
> >
> >
> > 1) It is still hard to read Figure 4. For example, it is always better to show the mask with a real manipulation, rather than manually drawing like 2nd row non-smile column. It is very hard to see what changes in the makeup-Lipstick macro column. Although it is good to see authors make Figure 4 more self-contained and specify the demonstration of interactive transfer, it is straightforward to ask why manually removing regions over the heatmap (mask), why not directly indicate the regions of interest to transfer directly from the RGB image?
> >
> > 2a) It is a little confusing that, on the one hand the authors say "the co-segmentation module is trained in a completely end-to-end manner, without any direct supervision", on the other "the used attribute label during training consists of `Smile’ and `Hair color’". Maybe how to train such a model is straightforward to other people, I don't understand how it is trained by reading the paper. Should I interpret training the cosegmentation as a weakly supervised learning that the attribute labels are provided during training, while the pixel-wise annotations are not provided?
> >
> > 2b) If interactive transfer is possible as discussed above in (1), a semantic segmentation should be also able to provide the region guide for style transfer on specified regions (by the segmentation model). Please explain clearly why such a segmentation method is incapable of targeted region transfer?
> >
> > 4) As for the pink colors, it should be improved further, even though the images are changed to alleviate this problem. I'm not sure if directly showing the heatmaps makes more sense, as overlaid images really effects the readability of the figure (as the pink color dominates the images denoted by 1-m, 1-m1 and 1-m2).
> >
> > 5) What is the meaning of AU 20? As readers (including me) may not be familiar with this dataset, missing the explanation of the index make it hard to follow.

---

> > > ### Author Response · Authors · 2018-12-19
> > > **Point-by-point response addressing the issues**
> > >
> > > ‘Updates on Fig. 4’
> > > Reflecting the reviewer’s comments, we significantly updated Fig. 4, as shown in http://123.108.168.4:5000/figure/page/4
> > > First, we included the resulting masks after user edits. Regarding makeup-lipstick example, the reason for a small difference is because the `lipstick’ and `makeup’ attributes, which are highly correlated, are difficult to disentangle,  and thus the noticeable difference is unlikely when applying only one of them. Instead, we replaced this example with the new combination of attributes (young and makeup), as seen in the last macro column of Fig. 4. Additionally, we strengthened the comparison results of LOMIT with other baseline methods, putting them separately in Fig. 6, which can be found at http://123.108.168.4:5000/figure/page/6
> > >
> > > ‘Topic of interaction’
> > > Directly indicating the region of interest in a given image may be a good alternative approach, but doing so from scratch may take much time especially when such a region has a complex shape, e.g., detailed hair regions. A user may have not have a clear idea on the region boundaries in the case of facial expressions. In this respect, partially editing the mask initially generated by the model can potentially take less time and give a better idea on where to edit, compared to the manual generation of the mask from scratch.
> > >
> > > ‘How co-segmentation module is trained’
> > > As the reviewer mentioned, we do not provide any groundtruth segmentation labels corresponding to the output of the co-segmentation module. Instead, our co-segmentation module is trained indirectly in an end-to-end manner using the image reconstruction loss (Section 4.2) and the auxiliary classifier loss (Section 4.4). That is, in order to preserve the original image as much as possible for the image reconstruction while still being properly classified as having the target attribute, the segmentation output is generated as the minimum possible region to clearly transfer the target attribute.
> > >
> > > ‘Applicability of semantic segmentation’
> > > One can definitely use a semantic segmentation approach in the place of a co-segmentation module in LOMIT, provided that pixel-level, pre-defined class labels are available. The co-segmentation approach we proposed in LOMIT can still work even when such labels are not available. We will include this discussion in the camera-ready version.
> > >
> > > ‘Issue of pink colors’
> > > To solve the issue, we changed the color scale of the overlaid mask to a grayscale one, as seen in http://123.108.168.4:5000/figure/page/23
> > >
> > > ‘Missing description about AU 20’
> > > AU 20 indicates a specific facial muscle corresponding to 'Lip stretcher,' which extends both corners of the mouth sideways. We will clarify this in the camera-ready version.

---

### Official Review · AnonReviewer2 · 2018-11-04
**Very structured but seemingly effective image 2 facial image translation.**

**Rating:** 6
**Confidence:** 4

**Review:**

The paper deals with image to image (of faces) translation solving two main typical issues: 1) the style information comes from the entire region of a given exemplar, collecting information from the background too, without properly isolating the face area; 2) the extracted style is applied to the entire region of the target image, even if some parts should be kept unchanged. The approach is called LOMIT, and is very elaborated, with source code which is available (possible infringement of the anonymity, Area Chair please check). In few words, LOMIT lies on a cosegmentation basis, which allows to find semantic correspondences between image regions of the exemplar and the source image. The correspondences are shown as a soft mask, where the user may decide to operate on some parts leaving unchanged the remaining (in the paper is shown for many alternatives: hair, eyes, mouth). Technically, the paper assembles other state of the art techniques,  (cosegmentation networks, adaptive instance normalization via highway networks) but it does it nicely. The major job in the paper lies in the regularization part, where the authors specify each of their adds in a proper way. Experiments are nice, since for one of the first times provide facial images which are pleasant to see. One thing I did not like were on the three set of final qualitative results, where gender change results in images which are obviously diverse wrt the source one, but after a while are not communicating any newer thing. Should have been better to explore other attributes combo.

---

> ### Author Response · Authors · 2018-11-27
> **The principle of performing a single-attribute translation**
>
> ‘Topic of single-attribute translation'
> To be concrete, LOMIT is trained for a multi-attribute translation (Gender and Facial Hair) while the output masks are interactively manipulated and forwarded into the networks to conduct a single-attribute translation (Gender or Facial Hair). We apologize for a confusing description of the figure. In order to clarify the figure, we have updated the figure and its description. Please refer to Fig. 4 and Subsection 5.3.
>
> ‘Issue of github page’
> We sincerely apologize for the mishap, and we have changed the URL to an anonymous one.
>
> We have uploaded an updated paper containing several improvements which we have highlighted.
> Finally, the demo website of LOMIT can be found at http://123.108.168.4:5000

---

### Meta-Review · Area_Chair1 · 2018-12-11

**Confidence:** 3
**Recommendation:** Reject

**Metareview:**

The paper received mixed ratings. The proposed idea is quite reasonable but also sounds somewhat incremental. While the idea of separating foreground/background is reasonable, it also limits the applicability of the proposed method (i.e., the method is only demonstrated on aligned face images). In addition, combining AdaIn with foreground mask is a reasonable idea but doesn’t sound groundbreakingly novel. The comparison against StarGAN looks quite anecdotal  and the proposed method seems to cause only hairstyle changes (but transfer with other attributes are not obvious). In addition, please refer to detailed reviewers’ comments for other concerns. Overall, it sounds like a good engineering paper that might be better fit to computer vision venue, but experimental validation seems somewhat preliminary and it’s unclear how much novel insight and general technical contributions that this work provides.